# Postoperative Delirium in Neurosurgical Patients: Recent Insights into the Pathogenesis

**DOI:** 10.3390/brainsci12101371

**Published:** 2022-10-09

**Authors:** Yinuo Xu, Qianquan Ma, Haiming Du, Chenlong Yang, Guozhong Lin

**Affiliations:** 1Department of Neurosurgery, Peking University Third Hospital, Beijing 100191, China; 2School of Basic Medical Sciences, Peking University Health Science Center, Beijing 100191, China; 3Center for Precision Neurosurgery and Oncology, Peking University Health Science Center, Beijing 100191, China; 4Department of Anesthesiology, Peking University Third Hospital, Beijing 100191, China; 5North America Medical Education Foundation, Union City, CA 94587, USA

**Keywords:** postoperative delirium, neurosurgery, cognitive dysfunction

## Abstract

Postoperative delirium (POD) is a complication characterized by disturbances in attention, awareness, and cognitive function that occur shortly after surgery or emergence from anesthesia. Since it occurs prevalently in neurosurgical patients and poses great threats to the well-being of patients, much emphasis is placed on POD in neurosurgical units. However, there are intricate theories about its pathogenesis and limited pharmacological interventions for POD. In this study, we review the recent insights into its pathogenesis, mainly based on studies within five years, and the five dominant pathological theories that account for the development of POD, with the intention of furthering our understanding and boosting its clinical management.

## 1. Introduction

Delirium is a temporary mental dysfunction characterized by confusion, anxiety, incoherent speech, hallucinations, and reduced awareness of the environment [1]. Postoperative delirium (POD) is a common complication in patients who undergo hospitalization and surgery and occurs most often in the hospital up to 1 week after surgery or until discharge [2,3]. Elderly patients undergoing surgery are at the highest risk of developing POD [4]. More than half of the elderly patients who underwent abdominal surgery were reported to experience POD. Moreover, approximately 54.9% of patients over 70 years of age underwent cardiac surgery and 6–56% of the general hospitalized population had symptoms of delirium [1,5]. Certain anesthetic interventions have been found to be associated with an increased risk of POD; notably, 31% of patients emerging from anesthesia had signs of delirium in the post-anesthesia care unit [6].

Patients with POD are under the threat of physically harming themselves without awareness. Patients may experience damage to the intravenous lines and tear the wound dressing. Furthermore, increasing evidence indicates that POD could severely affect multiple aspects of patient health. POD is associated with increased mortality in patients after transcatheter aortic valve implantation, with a 1-year survival rate decreasing from 85% to 68% [7]. In a retrospective study involving 1260 patients undergoing cardiac surgery, patients with POD experienced significantly more frequent postoperative complications, such as myocardial infarction, cerebrovascular accidents, respiratory complications, and infections [8]. POD also results in other complications, such as a prolonged hospital stay, delayed functional recovery, and increased morbidity [9].

The management of POD is also particularly important in neurosurgical units. Among those patients undergoing intracranial surgery, 4.2% had serious POD, which is directly associated with impaired neurological function and extended rehabilitation [10]. A meta-analysis involving 5589 patients revealed that the incidence of POD after intracranial surgery ranges from 12% to 26% due to variations in clinical characteristics and delirium assessment methods [11].

However, despite their prevalence and severity, pharmacological interventions with strong evidence for the prevention or management of delirium are sparse [12]. Even the promising agent dexmedetomidine fails to reduce postoperative delirium in patients recovering from cardiac surgery [13].

To address this harmful complication and improve the patient’s quality of life, we reviewed the major pathological mechanisms underlying the development of POD, the aim of which is to improve the prevention, remission, and medication of POD.

## 2. Risk Factors

Randomized results have shown that the incidence of POD is related to various perioperative risk factors. Knowledge of risk factors can help in clinical decision making and the identification of high-risk patients. The risk factors correlated with the onset of delirium, such as type of anesthesia, age, preoperative cognitive function, neurological function, and environmental factors, are summarized.

### 2.1. Anesthesia

The effect of the different types of anesthesia on POD is controversial. In a previous study, regional anesthesia and analgesia did not show any benefit with respect to POD compared with general anesthesia [14]. Recently, emerging evidence has been published to provide additional information on this topic. Regional anesthesia, such as caudal block, fascia iliac compartment block, and intravertebral anesthesia, is also mentioned to reduce the incidence of POD [15,16,17]. A recent study used topological data analysis to assess the phenotypic subgroups of delirium and suggested that elderly patients are more susceptible to POD and that this influence may be amplified by regional anesthesia [18]. In a recent study, data showed that regional anesthesia significantly reduced POD incidence and severity. The positive effect of regional anesthesia was especially reflected in pediatric patients rather than elderly patients on postoperative days 1–5 [19]. In neurosurgical operations, a longer duration of anesthesia is expected to induce POD due to the impairment of neurons. Patients with anesthesia for more than 4 h showed a dramatically higher incidence of POD. Interestingly, a history of anesthesia did not affect the occurrence of POD [20].

### 2.2. Age

POD is the most common complication following surgery in elderly patients. The incidence of POD has been estimated to range from 4% to 53% following fracture surgery in the elderly [21]. It is hypothesized that the increased risk of imbalance in cortical neurotransmitters or inflammatory responses contributes to delirium. In spinal surgery, patients younger than 73 years had a significantly lower incidence of delirium. Older age, low preoperative cognitive function, longer duration of surgery, and transfusion are important risk factors for POD [22]. Similarly, older age is also a critical intrinsic predictor of POD in those patients undergoing transcranial surgery. One study included patients aged 14–80 years who underwent brain tumor resection. The mean age of patients without and with POD was 47.1 ± 14.28 and 51.9 ± 12.66, respectively [20].

### 2.3. Cognitive Condition

Patients who undergo surgery and anesthesia are also at high risk of temporary or permanent cognitive impairment due to acute stress. Mounting evidence suggests that delirium may lead to permanent cognitive decline and dementia in some patients. Cognitive impairment was common (>50%) in surgical patients who developed POD, and the impairment persisted up to 1-year postoperatively [23]. Consistently, the cognitive state before surgery is a strong predictor of POD occurrence. Previous cognitive dysfunction, such as dementia or neurodegenerative status, greatly increases the vulnerability to POD [24,25]. In elderly patients with dementia, delirium is associated with increased rates of cognitive decline, admission to institutions, and mortality [23].

### 2.4. Intrinsic and Extrinsic Factors

The development of POD is multifactorial. Despite its main etiological factors, multiple intrinsic and extrinsic elements also promote the progression of POD. POD in neurosurgery patients can be induced due to hypothalamic syndrome, infection, electrolyte disturbances, fever, and profuse urination. A longer duration of ICU patients’ stay in the ICU was also associated with a higher incidence of POD [20]. In particular, cardiac arrest is an additional predisposing factor for POD because most survivors of cardiac arrest are treated in the ICU. A longer stay in the ICU has been hypothesized to induce POD. The occurrence of delirium prolongs the duration of the ICU and hospital stay and adversely affects functional outcomes [26].

Surgeons and anesthesiologists are devoting great effort to the prediction of the occurrence and the minimization of the risk of POD. The monitoring of processed electroencephalograms (EEGs) is considered an alternative method to predict the occurrence of POD. One moderate-quality evidence study indicated that EEG-optimized anesthesia could reduce the risk of POD in those patients aged ≥ 60 years who underwent non-cardiac or non-neurosurgical procedures [27]. However, a more advanced study did not agree with this conclusion. They found that in older adults who underwent major surgery, EEG-guided anesthetic administration, compared with usual care, did not decrease the incidence of POD [28].

## 3. Pathological Theories

Taking recent insights into comprehensive consideration, we conclude with five dominant pathological theories, as illustrated in Figure 1A, which may explain the occurrence and development of POD characterized by disturbances in attention, awareness, and cognition.

### 3.1. Neuroinflammation

Inflammation is inevitable after surgery as a protective response to injury. However, peripheral inflammation may trigger neuroinflammation, leading to the dysfunction of the central nervous system (CNS) and the subsequent neurobehavioral and cognitive symptoms of postoperative delirium. Additionally, inflammation from the periphery to the CNS starts with increased permeability of the blood–brain barrier (BBB).

The BBB is a highly regulated and maintained interface that separates the peripheral circulation from the CNS. A specific monolayer of endothelial cells, which forms the capillaries of the brain, is the main component of the BBB. Other components of the BBB anatomy include astrocytes, pericytes, neurons, and extracellular matrix [29,30].

Cellular injury caused by aseptic surgical trauma can induce the release of damage-associated molecular patterns (DAMPs) that activate the peripheral innate immune system [31]. S100A8 (Migration suppressor-associated protein-8 (MRP8)) is an important proinflammatory cytokine in many inflammatory conditions and is expressed in large quantities by activated neutrophils and monocytes [32,33]. As a main member of the DAMPs, S100A8 has been shown to promote the activation of Toll-like receptor 4 (TLR4) in macrophages and microglia [34,35,36]. In the TLR4 signaling pathway, MyD88 is an important activator of the NF-κB signaling pathway [37,38]. Previous studies have confirmed that S100A8 induced by surgery activates the TLR4/MyD88 pathway in mouse models [35]. A recent study in a rat model showed that S100A8/A9 binds to TLR4 and increases the expression of matrix metalloproteinases (MMPs), tumor necrosis factor α (TNF-α), and IL-6 (interleukin-6) through the NF-κB signaling pathways in nucleus pulposus cells, which contributes to inflammation-related pain [39].

The upregulation of TNF-α has been widely detected in postoperative patients [40,41]. TNF-α secreted by activated microglia cells binds to the TNF receptor (TNFR) on endothelial cells, subsequently triggering necroptosis through receptor-interacting protein kinase 1 (RIPK1), RIPK3, and mixed-lineage kinase domain-like pseudo kinase (MLKL), which disrupts the integrity of BBB and increases the permeability of BBB [42,43], as illustrated in Figure 1B. Moreover, TNF-α induces the release of MMP-9 from pericytes, resulting in increased endothelial permeability in BBB models in vitro [44].

The proinflammatory cytokine IL-6 is significantly upregulated in patients after orthopedic surgery [40,41,45]. In mouse models, IL-6 disrupts the BBB and promotes hippocampal inflammation through bone marrow-derived monocytes [46].

Upon infiltration into the brain parenchyma through the BBB, peripheral factors such as DAMPs and cytokines can trigger downstream neuroinflammatory responses.

Microglial cells, which account for 20% of the total glial cell population of the brain, are the main components of the posterior gray and white matter [47] and function to monitor the well-being of their environment and maintain homeostasis through innate defense mechanisms or specific immune reactions [48]. In a normal CNS environment, microglia are shut down with a scarce expression of many typical proteins on the surface of other tissue macrophages [49,50]. The activation states of microglial cells can be divided into M1 (classic activation) triggered by interferon-γ and M2 (alternative activation), which are mainly triggered by the Th2 cytokines IL-4 and IL-13 [51,52,53]. Activated microglial cells have long been considered the main source of proinflammatory factors such as cytokines, eicosanoids, complement factors, excitatory amino acids, reactive oxygen radicals, and nitric oxide [48,54]. Furthermore, owing to the microglial-secreted cytokines, the neurotoxic reactive subtype of astrocytes, astrocytes, can be induced by classically activated microglial cells, which contribute to the death of neurons and oligodendrocytes [55].

A recent study showed that neuronal dysfunction after traumatic brain injury, including disrupted neuronal homeostasis, reduced dendritic complexity, and defective compound action potential, can be attenuated by microglial elimination [56]. Moreover, in those patients undergoing abdominal surgery, microglial activation has been detected and associated with impairments in cognitive function [56]. Meanwhile, the endogenous mesencephalic astrocyte-derived neurotrophic factor has been shown to have positive effects on POD by inhibiting surgery-induced inflammation and microglial activation [57,58].

### 3.2. Oxidative Stress

Oxidative stress is an imbalance between the production of oxidants in cells and tissues and the particular biological processes that trigger the detoxification of these reactive products [59]. Under physiological and pathological conditions, cells can produce free radicals and reactive oxygen species (ROS) through the NADPH–oxidase system, xanthine oxidase, and mitochondrial electron transport chain [60].

The disruption of the BBB is a common cause of oxidative stress. Rat brain microvascular endothelial cells exposed to hydrogen peroxide (H_2_O_2_) at high concentrations display significant monolayer hyperpermeability with decreased cell viability and induced apoptosis [61]. It has been found that ROS can result in the BBB’s cytoskeleton rearrangement and redistribution and disappearance of tight junction (TJ) proteins, mediated by RhoA, PI3 kinase, and PKB signaling [62]. Occludin, a component of intercellular TJ protein complexes, moves away from TJ during oxidative stress [63]. Furthermore, toxic cell H_2_O_2_ concentrations cause occludin cleavage with the involvement of MMP-2 [64], and the upregulation of MMP-9 induces posttraumatic nerve and BBB injury, which may be partially mediated by Scube2 and SHH through the hedgehog pathway [65]. In turn, patients with increased disruption of the BBB are more vulnerable to neuronal injury induced by oxidative stress [66].

Oxidants have been found to increase the gating potential of the mitochondrial permeability transition pore (mPTP), resulting in hypersensitivity to Ca^2+^ activation during neuronal oxidative stress [67]. The high permeability of the mitochondrial membrane induced by mPTP dysfunction not only impairs the mitochondrial electron transport chain but also causes mitochondrial swelling, consequently leading to the over-release of ROS and neuronal necrosis [68]. Cyclosporine A, an inhibitor of mPTP opening that suppresses oxidative stress [69,70], was found to attenuate delirium-like behavior induced by anesthesia and surgery, with decreased ROS levels in the hippocampus of POD-like mice [71]. Meanwhile, as an innate protective mechanism to combat invading pathogens, macrophage cell lines, including microglia, can produce superoxide radicals and nitric oxide, resulting in the production and spread of peroxynitrite [72]. Peroxynitrite has been shown to induce neuronal apoptosis through the intracellular release of zinc and subsequent activation of p38 mitogen-activated protein kinase and caspase 3 [73].

Among those patients undergoing cardiac surgery, intraoperative oxidative damage has been found to contribute to postoperative delirium and neuronal injury [66,74]. Oxidative stress before surgery also makes elderly hip fracture patients vulnerable to POD [75]. Low baseline antioxidant capacity is independently associated with postoperative delirium development [76]. Additionally, increased levels of oxidative stress have been detected in patients with delirium [77].

### 3.3. Circadian Rhythm or Melatonin Dysregulation

Anesthesia is implicated in the modification and disruption of circadian rhythms after surgery [78,79]. Compared with general anesthesia, subarachnoid anesthesia is associated with less disruption of melatonin rhythm and sleep patterns and fewer POD occurrences in those patients undergoing hip fracture surgery [80]. Meanwhile, anesthesia mostly functions as a γ-aminobutyric acid (GABA) receptor antagonist and/or N-methyl-D-aspartate (NMDA) receptor agonist [78]. Additionally, neurons with the inhibitory neurotransmitter GABA [81] and GABAA receptors [82] account for the majority of the suprachiasmatic nucleus (SCN), which is the primary circadian pacemaker in mammals [83]. The activation of GABAA receptors by the agonist muscimol has been reported to directly affect pacemaker cells within the SCN and produce large phase delays in rats [83]. Moreover, NMDA receptors are expressed in the SCN [84], and its antagonist ketamine can induce phase changes in the secretion of melatonin and locomotor activity rhythms in a time-dependent manner [85]. Furthermore, higher levels of cortisol and inflammatory cytokines have been detected in patients undergoing morning surgery than in those undergoing afternoon surgery [86], supporting the theory that daytime anesthesia is a breach of circadian rhythms.

Circadian rhythm dysregulation, including sleep disorders, is a major symptom of POD [87] and promotes the development of delirium in hospitalized patients [88]. Sleep loss directly enhances the astrocytic phagocytosis of synaptic elements and promotes microglial activation, even without any noticeable signs of neuroinflammation [89]. The stimulation of α7 nicotinic acetylcholine receptor (α7-nAChR) has been found to attenuate cognitive decline, neuroinflammation, and oxidative stress [90], and reduced expression of α7-nAChR is detected in microglia and astrocytes after chronic sleep deprivation, accompanied by increased levels of proinflammatory factors and reduced levels of anti-inflammatory factors and antioxidant enzymes, which can be reversed by the α7-nAChR agonist PHA-543613 through the PI3/AKT/GSK-3β pathway [91].

Plasma melatonin functions as both a marker and regulator of endogenous circadian rhythms [92]. Melatonin has been found to combine with the NOTCH3 inhibitor DAPT, significantly enhancing DAPT’s inhibition of NF-κB/p65 translocation to the nucleus induced by IL-1β. Therefore, melatonin functions to protect the BBB from MMP-9 damage [93]. Moreover, in a rat model of intracerebral hemorrhage, melatonin reduced the number of apoptotic neurons, mPTP opening, BBB damage, inflammation, and oxidative stress [94].

Numerous studies have shown that melatonin can reduce the incidence of delirium in those patients undergoing surgery [95,96,97]. Recently, a meta-analysis involving 1712 participants found that melatonin significantly reduced the incidence of delirium, with a risk reduction of 49% in surgical patients [96]. In addition to melatonin, ramelteon, an FDA-approved melatonin receptor agonist, has also been shown to prevent delirium [97].

Studies have also suggested that sleep disruption before surgery predicts a higher risk of POD, since a greater share of wake after sleep onset during the night is detected before surgery and continues, even to a greater extent, postoperatively [98]. In addition to sleep disorders, the incoherence between circadian physical activity rhythms before and after hospitalization is associated with a higher risk of POD [99].

### 3.4. Older Age

Multiple studies have confirmed that older age is a risk factor for POD [100,101], with an increased risk of 3% at <65 years to 14% at 65–74 years and 36% at ≥75 years among patients with consecutive unselected acute medical admissions [102]. As mentioned above, microglial cells play a crucial role in the immune response of the CNS and the induction of neuroinflammation. With aging, senescent microglia may become dysfunctional, characterized by damaged structures and increased apoptosis [103,104]. Although the activity and phenotype of microglial cells are largely determined by the CX3CL1–CX3CR1 [105] and CD200–CD200R [106] axes, aging is associated with lower expression of CX3CR1 [107] and CD200 [108]. Additionally, with increased myelin degradation related to the aging process, microglia have been reported to be overwhelmed by the heavy workload to remove myelin fragments, leading to the senescence of microglia and immune dysfunction [109]. Microglia with an accumulation of lipid droplets have been identified in the aging brains of both mice and humans and are characterized by the dysfunction of phagocytosis and the overproduction of ROS and cytokines [110]. In contrast, the astrocytes mentioned above have a higher reactivity to microglia and neuroinflammation during the normal aging process [111]. Taken together, these findings showed that a proinflammatory state mediated in the aging brain contributes to greater vulnerability to injury.

### 3.5. Dysregulation of the Gut Microbiota

Anesthesia/surgery can cause different alterations in the gut microbiota of mice undergoing abdominal surgery, especially in POD mice, and treatment with *Lactobacillus* and probiotics can reverse anesthesia/surgery-induced changes and POD [112,113]. Patients undergoing cardiac surgery have been reported to have lower total bacterial counts and significantly higher fecal pH than preoperative levels [114]. This is similar to patients undergoing digestive surgery, characterized by an increased proportion of Gram-negative bacteria [115]. Many factors involved in surgery or anesthesia are associated with the dysregulation of the gut microbiota. Isoflurane and sevoflurane, commonly used as volatile anesthetics, have been reported to affect the motility and biofilm formation of bacteria, partially due to the interaction between volatile anesthetics and ion transporters [116]. Multiple antibiotics are associated with a higher risk of delirium [117,118], and the absence of gut microbial stimuli caused by broad-spectrum antibiotic treatment results in decreased levels of immune cells and cytokine production in the small intestine, colon, mesenteric lymph nodes, and spleen [119]. In addition to antibiotic drugs, in an extensive study with 1000 marketed drugs and 40 representative gut bacterial strains, 24% of non-antibiotic drugs with human targets were found to inhibit the growth of at least one strain of gut microbiota in vitro [120].

Altered β diversity and intestinal microbiota richness, as well as decreased levels of TJ proteins (ZO-1 and occludin), have been detected in the intestinal tract of mice with surgery-induced cognitive dysfunction [121]. Differential abundances of specific gut microbiota were detected between patients with and without POD after abdominal surgery, with a positive association between the phenotype of oxidative-stress-tolerant bacteria and POD [122]. A novel study suggested that oxidative stress and mitochondrial damage in microglia driven by the gut microbiota are probably the results of the metabolite N6-carboxymethyllysine [123]. Furthermore, alterations in certain bacteria associated with differential fecal metabolism of tryptophan, kynurenic acid, GABA, 2-indolecarboxylic acid, and glutamic acid in POCD mice suggest that the gut microbiota might contribute to cognitive dysfunction after surgery through neurotransmitter metabolism [124]. Taking the above findings together, we hypothesized that gut microbiota dysregulation may promote POD through inflammation worsened by a broken intestinal barrier, oxidative stress, and neurotransmitter disorders.

Consistent with the role that the gut microbiota might play in the development of POD, multiple studies have reported that probiotic treatment attenuates cognitive impairment in both animals and patients [125]. Treatment with the probiotic VSL3 inhibited neuronal apoptosis and reduced oxidative stress in POCD mice by upregulating the expression of microRNA-146a (miR-146a) and the inhibition of the B-cell translocation gene 2/Bcl-2-associated X protein (BTG2/Bax) axis [126]. ProBiotic-4, made up of *Bifidobacterium lactis*, *Lactobacillus casei*, *Bifidobacterium bifidum*, and *Lactobacillus acidophilus*, has been reported to significantly attenuate age-related cognitive dysfunction in mice with the inhibition of both TLR4-and RIG-I-mediated NF-κB signaling pathways and a reduction in IL-6 and TNF-α [127]. The administration of the prebiotic Bimuno (galactooligosaccharide mixture) significantly alleviated the cognitive decline induced by abdominal surgery under isoflurane anesthesia, accompanied by the reduced activation of microglia and expression of IL-6 [128]. Prebiotic (xylooligosaccharides (XOS)) intervention effectively attenuated surgery-induced cognitive dysfunction, as well as intestinal microbiota alterations, reduced inflammatory responses, and improved the integrity of the TJ barrier in the intestine and hippocampus [121].

## 4. Postoperative Delirium in Neurosurgical Patients

Patients who undergo neurosurgery are vulnerable to delirium. Among those patients who underwent brain tumor surgery, 4.2% were diagnosed with POD, which was associated with worse outcomes at hospital discharge [10]. POD was found to occur in 7% of glioblastoma patients and was associated with longer hospital stays, a lower probability of discharge home, and decreased survival [129]. A meta-analysis of 5589 patients revealed that the incidence of POD after intracranial surgery was 19%, ranging from 12% to 26%, due to variations in clinical characteristics and delirium assessment methods [11].

Cortical incision, the retraction of the brain lobes, and electrocoagulation are common procedures performed in neurosurgery. All of this can cause bleeding, edema, and direct injury to cerebral tissue [130]. Blood loss and tissue injury can activate immune cells and trigger the release of proinflammatory factors [131]. Additionally, more than 20% of patients with intracerebral hemorrhage, which may occur during neurosurgical procedures, develop a systemic inflammatory response syndrome [132]. Therefore, as mentioned above, common neurosurgical procedures can contribute to neuroinflammation and consequently promote the development of POD.

Studies have found that the activation of inflammatory cells after subarachnoid hemorrhages, such as microglia and neutrophils, can also generate free radicals [133]. Blood loss, caused by pre-existing diseases or neurosurgical procedures, can result in the accumulation of blood cells’ decomposition products, such as iron ions, heme, and thrombin, which consequently induce the production of free radicals [134]. Moreover, ischemia–reperfusion injury, commonly seen in neurosurgical patients with stroke, brain tumors, and subarachnoid hemorrhage, is mainly caused by oxidative stress. As a result, in addition to neuroinflammation, oxidative stress is also a major pathological factor involved in POD after neurosurgery.

Meanwhile, the specific use of anesthesia during neurosurgery and a longer duration of neurosurgical operation may contribute to the development of POD. Compared with remifentanil, the intravenous dose-dependent administration of fentanyl during neurosurgical procedures was found to have a strong association with POD [135]. In a study of 68,131 patients undergoing hip fracture repair, the risk of delirium increased with prolonged surgical duration, where every 30 min increase in the duration of surgery was associated with a 6% increase in the risk of delirium [136]. Since the detailed mechanisms underlying the association between these two factors and POD remain unknown, we presume that these pathological theories may apply, which deserves further research.

## 5. Conclusions

To accomplish this work, we gathered papers from the scientific database “PubMed”, using the keywords “postoperative delirium” and “neurosurgical”, separately and jointly. Meanwhile, we also gathered additional sources from the reference lists of some papers we found through database searches. Among these results, we selected, compared, synthesized, and summarized those related to POD’s pathogenesis. During this process, we preferred to choose novel and groundbreaking studies, mainly within 5 years, in order to reveal the recent insights into the pathogenesis of POD.

Postoperative delirium is a complication characterized by disturbances in attention, awareness, and cognitive function that occur shortly after surgery or emergence from anesthesia. Neuroinflammation, oxidative stress, circadian rhythm or melatonin dysregulation, advanced age, and gut microbiota dysregulation have been shown to interact with each other and play crucial roles in the development of POD. However, the detailed and specific mechanisms underlying POD require further research and analyses. Furthermore, while most studies focus on the pathological factors that occur during the operation, personal characteristics that existed before the operation are emphasized and are worth future attention.

Given that there is a lack of pharmaceutical intervention to prevent and manage POD and most drug discoveries in the past focused on antipsychotics, acetylcholinesterase inhibitors, steroids, and statins, we propose that the gut microbiota might be a therapeutic target, and probiotics might be a promising direction for pharmaceutical development.

## Figures and Tables

**Figure 1 brainsci-12-01371-f001:**
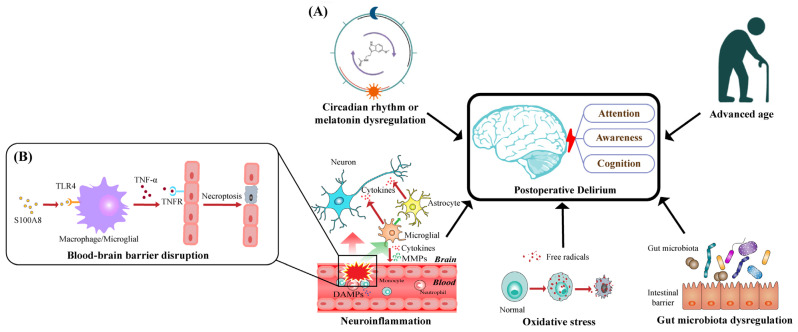
Schematic diagram of pathological mechanisms underlying postoperative delirium: (**A**) five dominant pathological theories that may account for the occurrence and development of POD characterized by disturbances in attention, awareness, and cognition; (**B**) S100A8, as a main member of DAMPs, promotes the activation of TLR4 in macrophages and microglia and then increases the expression of TNF-α; TNF-α will bind to TNFR on endothelial cells, subsequently triggering their necroptosis, which disrupts BBB’s integrity and increases BBB’s permeability.

## Data Availability

Not applicable.

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
