# Peer review of "Postoperative Delirium in Neurosurgical Patients: Recent Insights into the Pathogenesis"

_brainsci, 2022, doi:10.3390/brainsci12101371_

Round 1
Reviewer 1 Report
Dear Authors, many thanks for the opportunity to read your narrative paper
I think it deserves several improvements
Please try to explain how you choose your references (which criteria?)
Please discuss better
1) anaesthesia role is controversial
2) The role of neuromoniotring intraoperative
3) The role of preoperative level of hemoglobin
4) Type of neruosurgery (only head-neck surgery? or also spinal surgery? Traumatic brain injury? What about CAM-ICU
What about cognitive assessment?
5) The role of pain
6) enviromental factors
I give to you some references: these are suggestions interesting to read: do what you want.
Wildes, T.S.; Mickle, A.M.; Abdallah, A.B.; Maybrier, H.R.; Oberhaus, J.; Budelier, T.P.; Kronzer, A.; McKinnon, S.L.; Park, D.; Torres, B.A.; et al. Effect of electroencephalography-guided anesthetic administration on postoperative delirium among older adults undergoing major surgery the engages randomized clinical trial. JAMA-J. Am. Med. Assoc. 2019, 321, 473–483
Punjasawadwong, Y.; Chau-in, W.; Laopaiboon, M.; Punjasawadwong, S.; Pin-on, P. Processed electroencephalogram and evoked potential techniques for amelioration of postoperative delirium and cognitive dysfunction following non-cardiac and non-neurosurgical procedures in adults. Cochrane Database Syst. Rev. 2018, 2018, 1465–1858.
Reviewer 2 Report
Thank you for the opportunity to review.
The work is very good, interesting and innovative. More similar work is needed on delirium in various fields of medicine.
I present a few small remarks below:
- In the ABSTRACT section, include information consistent with the guidelines, such as the purpose of the study, methods;
- How were the articles for the review selected?
- What were the criteria for inclusion of works in the review?
- I am asking for the inclusion of a few new items from the literature:
https://www.ncbi.nlm.nih.gov/pmc/articles/PMC9320343/
Congratulations to the authors and good luck.
Reviewer
Reviewer 3 Report
Although the papers is of interest, there are some suggestions which can significantly improved the manuscript.
1. The Authors nicely presented possible mechanisms underlying POD. However, since quite a lot of papers indicate that anaesthesia and furthe post-operative analgesia are the main factors that may induce POD, I suggest to provide additional information regarding this issue.
2. Since there is a lot of important information on BBB disruption as a consequence of a surgery or - as provided by the Authors BBB can be easily crossed by various inflammation-related factors, in my opinion it would be nice if the Authors could try to present/organize such as a figure or maybe a table.
3. The Authors have higlighted several pathological mechanisms underlying postoperative delirium, including age. What about polypharmacy, or the occurence of a disease? Please extend the manuscript by focusing also on the above-mentioned.
4. In my opinion, the Authors devoted too little space to present information about the topic and - de facto - the title of the work. In fact, there is not enough information on "POD in neurosurgical patients.
5. There are some grammar and punctuation mistakes.
6. Based on the conclusion section presented by the Authors, the main text of the manuscript should be much extended
Round 2
Reviewer 1 Report
the paper has been improved
Plagiarism checked with grammarly: less than 4%
Very good
Reviewer 3 Report
The paper has now been greatly improved. Therefore, in my opinion the paper is now accepted for publication.